# Direct protein-lipid interactions shape the conformational landscape of secondary transporters

Chloe Martens[1], Mrinal Shekhar[2], Antoni J. Borysik[1], Andy M. Lau[1], Eamonn Reading [1], Emad Tajkhorshid [2], Paula J. Booth[1] & Argyris Politis[1]

Secondary transporters undergo structural rearrangements to catalyze substrate translocation across the cell membrane – yet how such conformational changes happen within a lipid environment remains poorly understood. Here, we combine hydrogen-deuterium exchange mass spectrometry (HDX-MS) with molecular dynamics (MD) simulations to understand how lipids regulate the conformational dynamics of secondary transporters at the molecular level. Using the homologous transporters XylE, LacY and GlpT from *Escherichia coli* as model systems, we discover that conserved networks of charged residues act as molecular switches that drive the conformational transition between different states. We reveal that these molecular switches are regulated by interactions with surrounding phospholipids and show that phosphatidylethanolamine interferes with the formation of the conserved networks and favors an inward-facing state. Overall, this work provides insights into the importance of lipids in shaping the conformational landscape of an important class of transporters.

[1] Department of Chemistry, King's College London, 7 Trinity Street, London SE1 1DB, UK. [2] Center for Biophysics and Quantitative Biology, Department of Biochemistry, NIH Center for Macromolecular Modeling and Bioinformatics, Beckman Institute for Advanced Science and Technology, University of Illinois at Urbana-Champaign 405N. Mathews Ave., Urbana, Illinois 61801, USA. Correspondence and requests for materials should be addressed to A.P. (email: argyris.politis@kcl.ac.uk)

Secondary membrane transporters play crucial roles in maintaining adequate conditions for life by catalyzing uphill transport of biomolecules through the biological membrane, using energy stored in transmembrane ions gradients. The majority of known secondary transporters are grouped in the Major Facilitator Superfamily (MFS), which represents the largest evolutionary related family of transporters[1]. This family comprises many disease-related human transporters[2]. For example, the GLUT sugar transporters are involved in cancer metabolism[3] and malfunction of the glucose-6-phosphate transporter (G6PT) is associated with glycogen storage disease[4]. Despite functional diversity, the architecture of MFS transporters is remarkably well conserved, which suggests that key mechanistic features are maintained within the family[5]. These proteins are dynamic entities, coupling opening on one side of the membrane with closing on the opposite side to provide alternate access to a central substrate binding site[6]. The typical structural fold of the transporters contains 12 transmembrane helices (TMs) arranged in two pseudo-symmetrical six-helical bundles, the N-lobes and C-lobes, rocking back and forth to allow substrate translocation[7]. Recently, high-resolution crystal structures have provided invaluable molecular insights into the alternating-access mechanism of MFS transporters by unveiling various conformational states ranging from outward-facing (OF) to inward-facing (IF)[8–10].

Despite the wealth of information offered by atomic-resolution structures, they can only give very limited information on the important role that the lipid environment plays in the molecular mechanism of transport. Recent studies have shown that the biophysical properties of the membrane and lipid–protein interactions play a crucial role in modulating the function[11,12], structure[13], stability[14–16], oligomeric state,[17,18] and conformational dynamics[19,20] of transporters. Furthermore, the static structural snapshots fail to report the dynamic interconversion between different conformers, which underpins the transport cycle. Thus, a method that interrogates the conformational dynamics within a lipid environment is required to gain fundamental insights into the mechanism of membrane transport.

Recently, hydrogen–deuterium exchange mass spectrometry (HDX-MS) has emerged as a powerful technique to monitor the conformational dynamics[21] of membrane proteins in various membrane mimics such as detergent micelles[22,23], bicelles[24], native lipid nanodiscs (SMALPs),[25] and artificial lipid nanodiscs[26] as well as liposomes[27]. HDX reports the exchange of labile amide hydrogens on the protein backbone in the presence of bulk deuterium[28]. When coupled to enzymatic digestion and followed by liquid-chromatography mass spectrometry (LC-MS) analysis of the resulting peptides, this technique allows deuterium uptake to be mapped at a peptide level of resolution. The most insightful information stems from differential measurements of relative deuterium uptake, obtained by comparing two different protein states. Significant differences in deuterium uptake between two distinct conditions (e.g. mutant and wild-type protein) are mapped onto a 3D protein structure of allowing visualization of the changes in structural dynamics[29]. A major advantage of this methodology is that it enables conformational characterization of proteins in solution without requiring covalent modification of the target, circumventing labeling issues that arise in many biophysical studies. A recent study on the neurotransmitter LeuT combined HDX-MS in nanodiscs with molecular dynamics (MD) simulations to identify dynamic changes in key structural elements demonstrating the successful application of this method to membrane proteins[30]. However, no study yet has used HDX-MS to study the role that lipids have on the conformational dynamics of transporters.

Here, we describe the systematic investigation of the conformational landscape of three well-characterized transporters:

lactose permease LacY, xylose transporter XylE, both symporters, and glycerol-3-phosphate antiporter GlpT, all from *Escherichia coli*. We chose these three targets due to the availability of their crystal structures[9,31,32] and the potential insights our results could provide into the mechanism of their mammalian counterparts. GlpT and XylE are bacterial homologs of the mammalian transporters G6PT and GLUT1, respectively, LacY is selected as it represents the most extensively characterized secondary transporter to date[33]. Our HDX-MS experiments in detergent micelles identify cytoplasmic charge networks of amino acids that act as molecular switches controlling the conformational equilibrium, through a mechanism conserved across all three transporters. Combining MD simulations and HDX-MS experiments of XylE and LacY embedded in nanodiscs of different lipid compositions, we show that phosphatidylethanolamine (PE) modulates the conformational equilibrium between OF and IF states through interactions with the identified networks of charged residues. Overall, this work suggests a model of secondary transport in which interactions between lipids and protein at conserved charge networks control the conformational equilibrium.

## Results

**HDX-MS reports on conformational changes of transporters.** An important prerequisite to our mechanistic investigation was to ensure that HDX-MS can consistently report on changes in the conformational equilibrium of different MFS transporters. To validate our methodology, we shifted the conformational equilibrium toward OF conformers by introducing a bulky tryptophan in place of a conserved glycine on the extracellular side between helix 2 and 11 (Fig.1a and Supplementary Fig. 1). This approach was successfully utilized in the first high-resolution structure determination of LacY in an OF conformation[34]. We measured the difference in relative deuterium uptake (ΔHDX) between the mutants LacY G46W, XylE G58W, and GlpT G66W and the respective wild-type (WT) transporter in detergent micelles (Fig. 1b and Supplementary Figs. 12, 13). Only the peptides displaying a significant ΔHDX (99% CI) were kept for downstream analysis (see Methods). By mapping the ΔHDX values on the crystal structure of each transporter, we observe that peptides from the intracellular and the extracellular sides display opposite deuterium uptake profiles. Regions on the extracellular side of all three mutants take up more deuterium compared with the wild-type, whereas regions on the intracellular side are comparatively protected from deuterium exchange. This pattern is consistent with a shift in the conformational equilibrium toward an OF state and shows that HDX-MS can report on changes in the conformational equilibrium of MFS transporters within the framework of the alternating-access mechanism.

**Conserved charge networks control the conformational shift.** We then set out to identify molecular switches regulating the conformational transition between IF and OF conformations. The most conserved sequence motif of the MFS family, originally termed motif A[35], is the central point of charge-relay networks present on the intracellular side (Fig. 2a and Supplementary Figs. 2, 3). Previous structural and biophysical studies have suggested that these charge networks play a role in stabilizing the OF conformation[36–38]. To understand how this motif regulates transporter dynamics, we disrupted the charge-relay networks while minimizing steric perturbation by introducing conservative mutations of its acidic residues. We produced the following mutants: D68N and E139Q for LacY, D337N, E397Q, and E153Q for XylE, and D314N and E374Q for GlpT, and subjected them to differential HDX-MS measurements using the respective WT transporter as the control. Interestingly, our ΔHDX maps show

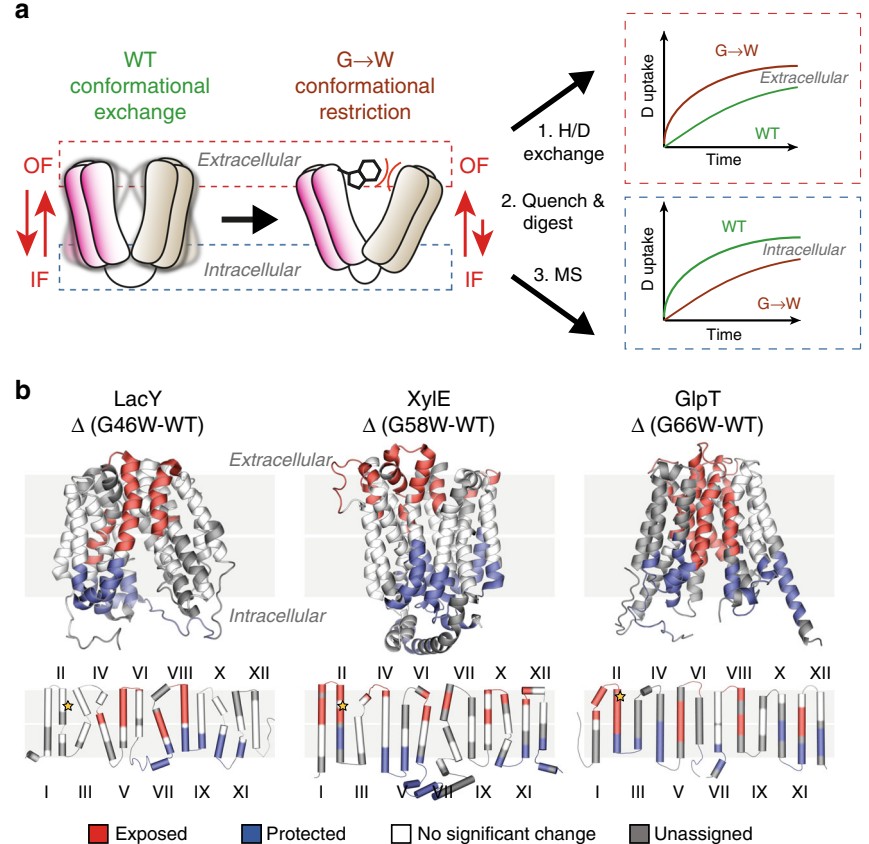

**Fig. 1** HDX-MS reports on changes in the conformational equilibrium between IF and OF states. G to W mutants were used to highlight shifts in the conformational equilibrium. **a** Introduction of a tryptophan at the extracellular end of helix 2 sterically prevents closing. After deuteration, enzymatic digestion, and identification of the peptides by MS the mass uptake of the WT and mutants is compared. Peptides located on the intracellular side will be on average more deuterated than the peptides located on the extracellular side for the mutant (top graph) while the opposite pattern will be observed for the WT (bottom graph). **b** Differential deuterium uptake pattern (ΔHDX) mapped onto the 3D and topological structure of LacY (PDB: 2V8N), GlpT (PDB: 1PW4), and XylE (PDB: 4GBY). Red and blue colored regions indicate segments containing peptides with a positive ΔHDX (red—more deuteration) or negative ΔHDX (blue—less deuteration), respectively; white regions indicate that no significant ΔHDX is observed ($P \leq 0.01$), and gray indicates regions where peptides were not obtained for both the mutant and the WT conditions. The yellow star indicates the location of the point mutation. All measurements were performed in triplicates. The ΔHDX datasets are presented as Woods plots in supplementary Figs.12 and 13

that all mutations trigger increased deuterium uptake on the intracellular side coupled to decreased uptake on the extracellular side (Fig. 2b–d). These patterns indicate that the IF conformation is favored in the presence of these mutations. These results strongly suggest that the conserved networks on the intracellular side play a role as conformational molecular switches.

Another characteristic feature of secondary MFS transporters is the presence of conserved charged residues within the transmembrane region (Supplementary Fig. 3). Previous functional characterization of the residues D27 for XylE[9], E325 for LacY,[33] and E299 for GlpT[39] showed that these are essential for transport but not for substrate binding, which suggests that they are likely to play a role in structural rearrangements enabling alternating-access. We mutated each carboxylate residue into its amine equivalent (i.e D27N for XylE, E325Q for LacY, and E299Q for GlpT) and monitored the changes in deuterium uptake compared with the respective WT transporter using differential HDX-MS. In contrast to our previous results, the three transporters displayed different conformational responses (Supplementary Fig. 4). The ΔHDX map of GlpT suggests that the perturbation of the salt-bridge between E299 and K46 caused a shift toward OF conformation (Supplementary Fig. 4a). ΔHDX maps of XylE and LacY indicate a closing of the extracellular side which is not

coupled to an opening on the intracellular side (Supplementary Fig. 4b, c). Such independent rearrangements at both sides of the transporter has already been suggested for LacY[40] and may indicate a mechanism shared by other MFS transporters.

Together, these results underline the complexity of the molecular mechanisms involved in the conformational changes over the course of the transport cycle. The disruption of the extracellular salt-bridges causes a variety of structural responses, which may reflect the differences in energetics between the antiporter GlpT and the symporters XylE and LacY. In contrast, the disruption of the cytoplasmic charge-relay networks consistently opens the intracellular side, independent of the transporter type or specific function.

**Lipids modulate the conformational equilibrium**. Having established that disruption of conserved cytoplasmic charge networks triggers a conformational transition toward the IF state, we investigated the role of specific lipid–protein interactions in regulating this transition. A previous study on the MFS multidrug transporter LmrP suggested that a direct interaction between PE and a conserved cytoplasmic network facilitated the conformational transition between IF and OF states[19].

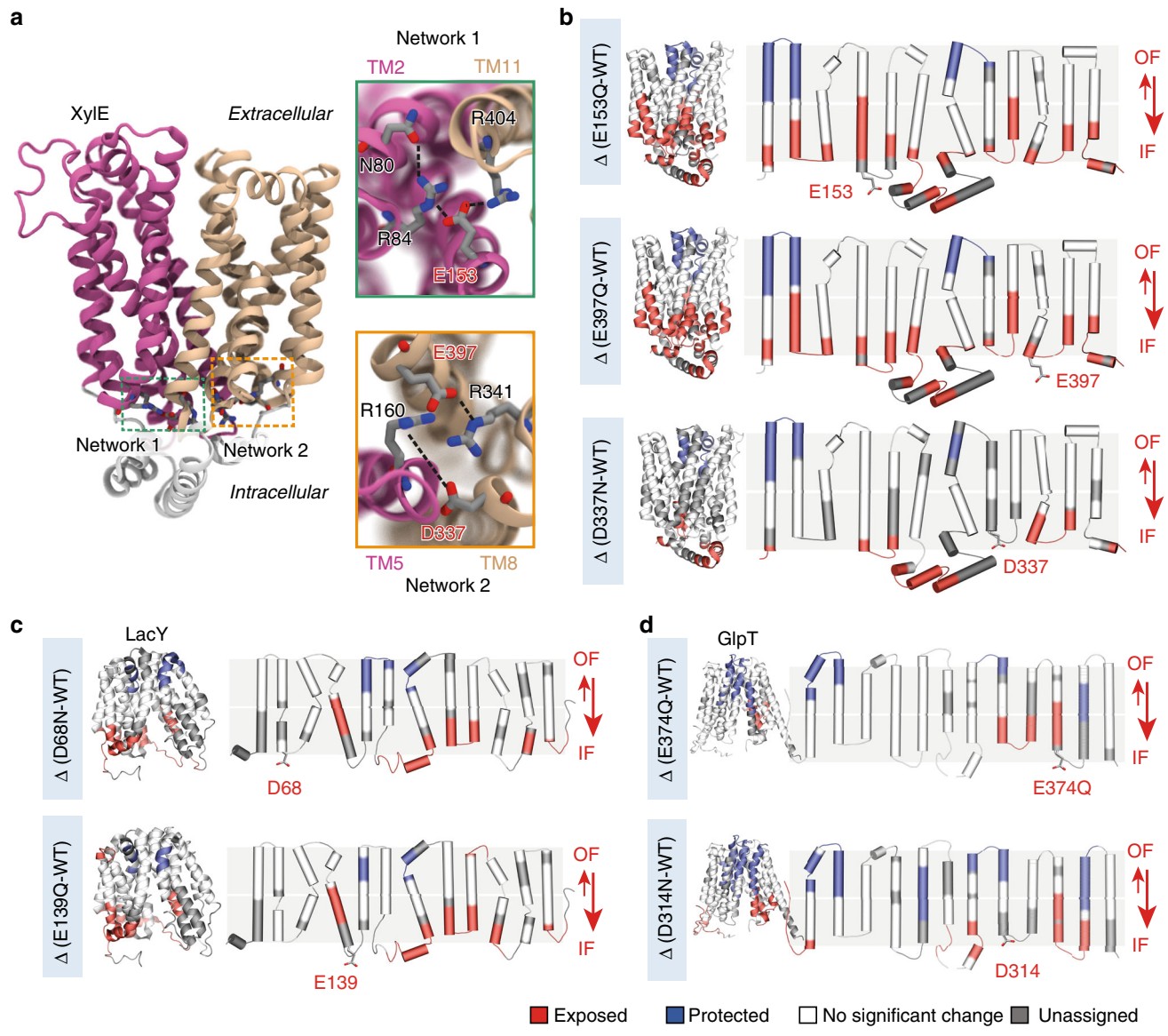

**Fig. 2** Conserved charge-relay networks control intracellular opening. Disruption of the networks by conservative mutations (D to N and E to Q) shifts the conformational equilibrium toward the IF state. **a** Two charge-relay networks of conserved residues stabilize the OF state of XylE. The N- and C-lobes are colored pink and tan, respectively. Close-up of networks 1 and 2—highlighted in green and orange, respectively—show the connection between the two lobes and highlight in red the mutated acidic residues. **b** ΔHDX between the mutants E153Q, E397Q, and D337N vs. WT mapped onto the 3D and topological structure of XylE. **c** ΔHDX of the mutants D68N, and E139Q vs. WT mapped onto the 3D and topological structure of LacY. **d** ΔHDX of the mutants E374Q and D314N mapped onto the 3D and topological structure of GlpT. All measurements were performed in triplicates on two biological replicates. The ΔHDX datasets are presented as Woods plots in supplementary Figs. 12 and 13

Building upon this knowledge, we characterized the conformational shifts of XylE in nanoscale soluble phospholipid bilayers (nanodiscs) composed of two different lipid environments: (a) PE, phosphatidylglycerol (PG), and tetraoleyl cardiolipin (CL), or (b) phosphatidylcholine (PC), PG, and CL, both in a 7:2:1 ratio. The first resembles the native lipid composition of the *E. coli* membrane[41] and the second contains the non-native PC lipid and is used as a control. We utilized 1,2-dioleoyl-sn-glycerol lipids in each nanodisc because they are among the most abundant lipid types in bacterial membranes and they all have transition temperatures below 0 °C, allowing for efficient reconstitution. Following careful optimization of the HDX workflow (Fig. 3a and Methods), we obtained a high level of sequence coverage in nanodiscs ranging from 76% up to 85% (Supplementary Fig. 5).

To demonstrate that the protein is functional when inserted in the nanodiscs, we mapped ΔHDX in the absence and presence of the substrate xylose, and repeated the experiment in detergent micelles. In both cases, the presence of substrate shifted the conformational equilibrium toward the OF state (Supplementary Fig. 6). The associated conformational changes confirmed xylose binding and offered a good indication that the protein is folded and reconstituted. As additional controls, we used thin layer chromatography to ensure that the nanodisc reconstitution procedure does not select a specific lipid species (Supplementary Fig. 7) and native MS to assess the absence of detergent trace in the final nanodiscs samples (Supplementary Fig. 8).

Interestingly, we observe a decrease in deuterium uptake on the extracellular side in the PE nanodiscs compared with the PC

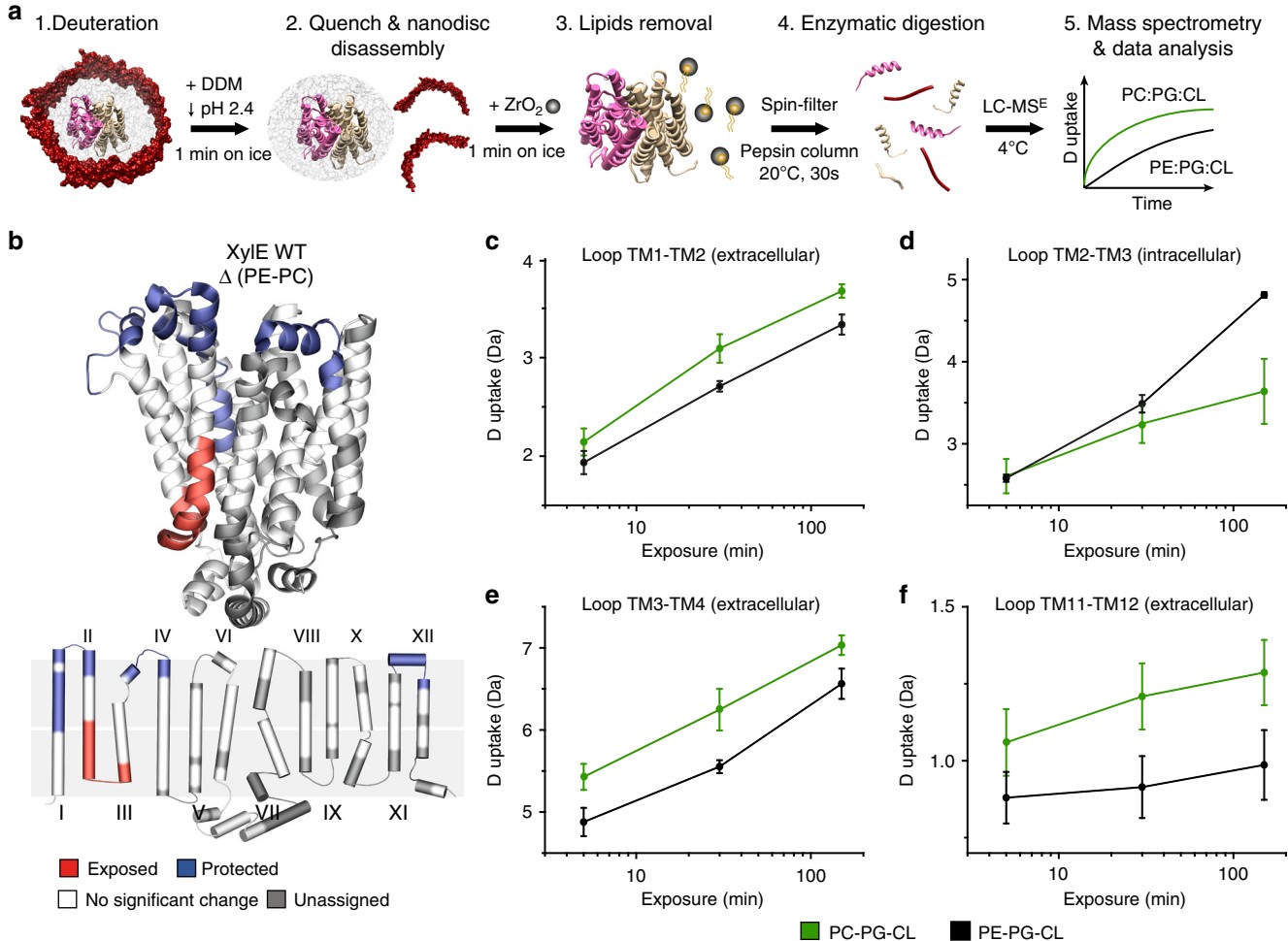

**Fig. 3** Lipid–protein interactions regulate the conformational equilibrium. ΔHDX of XylE WT in PE vs. PC nanodiscs shows that PE promotes the IF conformation. **a** Optimized experimental procedure for HDX-MS of a transporter in nanodiscs of different lipid compositions. Following deuteration at specific time points, the exchange reaction is quenched and the nanodiscs disassembled with detergent DDM. Lipids are removed by adsorption, before enzymatic digestion and peptides identification by MS. The mass uptake of the protein in both lipid environments is then compared. **b** ΔHDX of WT XylE in DOPE-PG-CL nanodiscs (native-like) minus DOPC-PG-CL (control) nanodiscs mapped on the PDB structure and the topological representation. **c** Representative deuterium uptake plots for peptide 28–38, (**d**) 70–88, (**e**) 108–122, (**f**) 430–434 in DOPC-PG-CL nanodiscs (green) and DOPE-PG-CL nanodiscs (black). Standard deviations for each time point are plotted as error bars ($n = 3$). All measurements were performed in triplicates

nanodiscs. This is coupled to an increase in deuterium uptake on the intracellular side (Fig. 3b). Specifically, the loops between TM1 and TM2, TM3 and TM4, and TM11 and TM12 on the extracellular side and the loop between TM2 and TM3 on the intracellular side displayed significant ΔHDX between the two lipid environments (Fig. 3c–f). This ΔHDX profile suggests that the presence of PE favors the opening on the intracellular side and closing on the extracellular side, promoting a shift toward an IF conformation. To test the generality of such result, we carried out an identical experiment using LacY. Similar to XylE, we observed an increase in deuterium uptake on the intracellular side, coupled to a slight decrease on the extracellular side in the PE nanodiscs compared with the PC nanodiscs (Supplementary Fig. 9). These results indicate that the composition of the phospholipid headgroup significantly affects the conformational equilibrium of both secondary transporters.

**MD simulations reveal direct lipid–protein interactions.** Having established that a change in the lipid environment modulates the conformational dynamics of the transporters XylE and LacY, we set out to identify the origin of such conformational regulation

at a molecular level. We carried out all-atom MD simulations of each transporter embedded in lipid bilayers of identical composition to the nanodiscs used in our HDX-MS experiments. We performed two sets of simulations of 500 ns each (Methods), starting from the crystal structure of XylE and LacY resolved in the inward-open conformation (PDB 4JA4 and 2CFQ, respectively) (Fig. 4a and Supplementary Fig. 10).

Interestingly, the MD simulations predicted direct interactions between the lipid bilayer and cytoplasmic networks identified as molecular switches (Fig. 4b, c). The simulations of XylE in PE:PG:CL bilayers consistently showed PE interacting with the networks, either by direct interactions of the PE headgroup with the acidic residues E153 (Fig. 4b, c and Supplementary Movie 1) and D337 (Fig. 4c) or by steric interferences with the whole networks (Fig. 4c). In contrast, no specific or non-specific interactions between the protein and PC were observed (Fig. 4c). Similar results were obtained for LacY, where direct interactions between PE and the acidic residue E139 were observed in PE but not in PC bilayers (Supplementary Fig. 10 and Supplementary Movie 2). Furthermore, the simulations of XylE in PC:PG:CL bilayers predicted a closing of the intracellular side of the protein. This

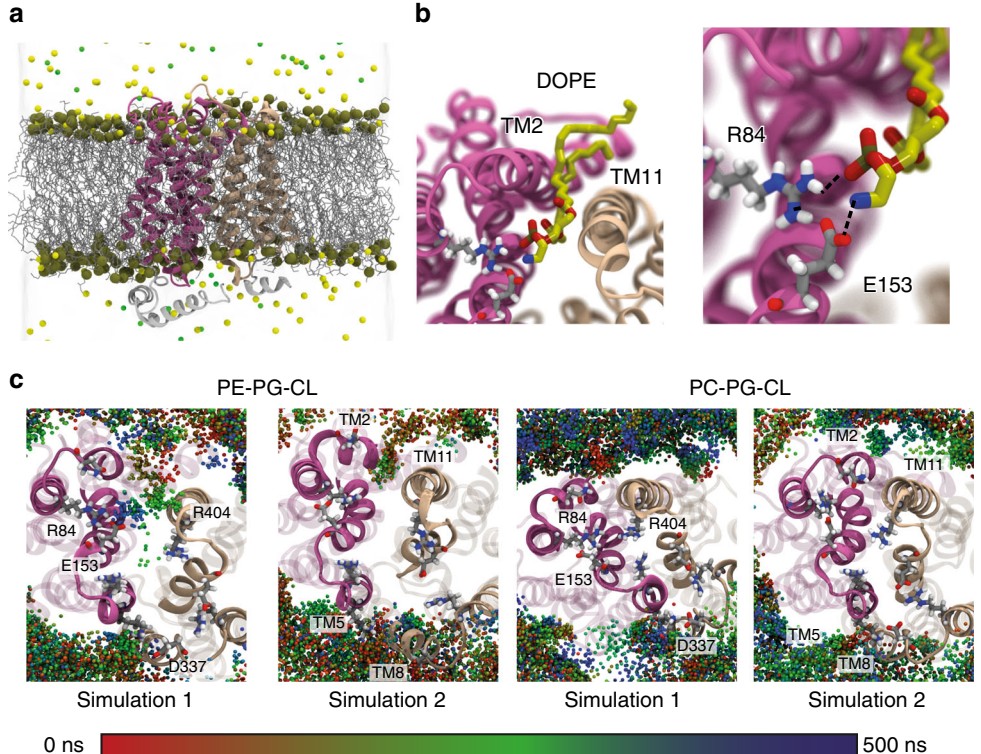

**Fig. 4** MD simulations reveal that PE–protein interactions perturb the charge-relay networks. MD simulations predict specific interactions between the PE headgroup and charged residues of networks 1 and 2 in XylE. **a** Simulation box. Representative snapshot of XylE (PDB: 4JA4) embedded in an explicit lipid bilayer and water box with 100 mM NaCl. Lipid molecules are shown in line representation colored in silver, phosphorus atom of the headgroup represented by orange spheres. Yellow and green spheres represent Na⁺ and Cl⁻ ions, respectively. **b** Representative snapshot of the close-up of network 1 interacting with the PE headgroup of the phospholipid. Polar interactions with R84 and E153 prevent network formation and steric hindrance prevents contacts of the TM2 and TM11. **c** Location of the phosphorus atoms of the PE/PC lipid molecule in two 500 ns trajectories in PE:PG:CL bilayers (left) and PC:PG:CL bilayers (right). The spheres are color coded: red at t = 0 ns and blue at t = 500 ns. In both simulations 1 and 2, PE lipids wedge between the helices (more so in simulation 1), but no lipid–protein interaction is observed in PC bilayers. In simulation 1, direct interactions with residues R84, E153 and R404 of network 1 are observed

conformational change is initiated by the inward motion of TM2 and TM3 toward TM10 and TM11 (Fig. 5a). Such a conformational change was not observed in PE-based lipid bilayers. Specifically, in the last 200 ns of both simulations, we observed that the intracellular gates converge to a significantly shorter 9 Å distance in the presence of PC but stay open at distance of ~13 Å in the presence of PE (Fig. 5b, c). This is in line with our HDX-MS observations showing that PE lipids shift the conformational equilibrium toward an IF state. These observations suggest that the lipid-induced conformational shift results from direct interaction of the conserved cytoplasmic networks with PE.

We speculated that such an effect would be abolished or decreased if the charge networks are already perturbed or disrupted. To test this hypothesis, we reconstituted the XylE mutants E153Q (disruption of network 1) and E397Q (disruption of network 2) into nanodiscs containing either PE or PC and subjected them to differential HDX-MS. The mutant XylE E153Q did not show any significant ΔHDX between the two lipid environments (Fig. 6a), confirming that the perturbation of network 1 nullifies the PE-induced conformational shift. The mutant XylE E397Q displayed no significant ΔHDX, with the only exception of the TM1-TM2 extracellular loop, more exposed in PC-based nanodiscs (Fig. 6b). These experiments strongly suggest that PE favors inward opening by perturbing the cytoplasmic charge networks we evidenced. To further confirm that the conformational changes were caused by direct lipid–protein interaction and not differences in the curvature

properties of PE and PC, we performed differential HDX-MS measurements of XylE in nanodiscs composed of DOPE with one methyl group on the amine headgroup (DOPE(Me)₁:DOPG:CL) and in DOPC:DOPG:CL lipid nanodiscs. According to previous work[42], the curvature properties of DOPE decreases linearly with the addition of each methyl group on the amine. Hence, the curvature properties of the DOPE(Me)1 nanodiscs should be comparatively closer to the DOPE than the DOPC nanodiscs. If the observed shift in the conformational equilibrium is caused by a difference in curvature properties, such shift should still be observed. However, our HDX-MS experiments show that there is no significant difference in uptake between the DOPC and DOPE (Me)₁ nanodiscs, which indicates that their conformational landscape is identical (Supplementary Fig. 11). Overall, the combination of ΔHDX experiments and MD simulations in two lipid environments shows that the conformational dynamics of secondary transporters are controlled by specific lipid–protein interactions.

## Discussion

In this work, we identify conformational roles for the most conserved motif of the MFS superfamily, occurring through specific lipid–protein interactions. Furthermore, we show that ΔHDX measurements in nanodiscs of different lipid compositions can be used to understand how specific lipid environments fine-tune the conformational dynamics of transporters.

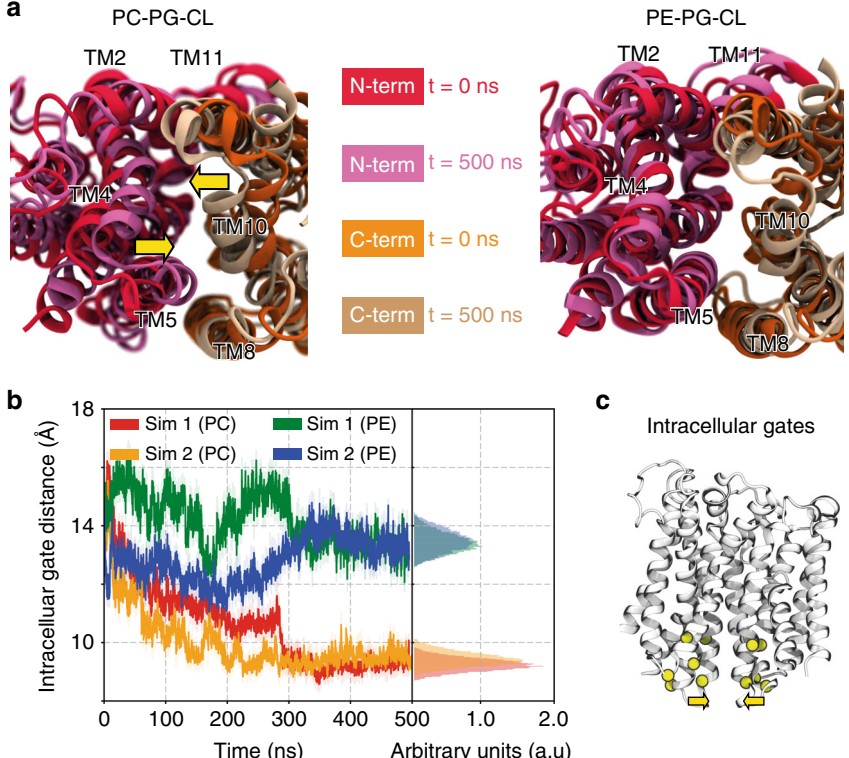

**Fig. 5** Closing of the intracellular side of XylE in PC:PG:CL bilayers. MD simulations predict different conformational changes of XylE in PC vs. PE bilayers. **a** Representative snapshots at the beginning (t = 0 ns) and end of the simulation (t = 500 ns) show a closing of the intracellular side in PC nanodisc (left) compared to PE nanodisc (right). **b** Time trace showing the intracellular gate distance over simulation time. PC simulation sets are coloured orange and red, while the PE simulation sets are coloured green and blue. The inset on the right depicts the normalized distribution of the intracellular gate distance for the last 200 ns of the trajectory. **c** Snapshot describing the definition of the intracellular gates in XylE. The intracellular gating distance is defined as the center of mass (COM) distance between the two groups of Ca-Ca residues: group1 residues (75–80, 149–154, 160–16) and group2 residues (332–337, 391–397, 404–410)

A number of independent studies have previously proposed that motif A integrity is crucial to stabilize the OF conformation of MFS transporters through conserved charge-relay networks[37,38,43]. We demonstrate that the disruption of these cytoplasmic charge networks by mutagenesis or the presence of specific lipids systematically causes a shift toward the IF state (Fig. 2b–d), a finding consonant with the proposed structural role of the A-like motifs. The conservation of A-like motifs in MFS transporters from a variety of organisms suggests that the disruption-reformation of these networks is likely to be a conserved mechanism driving conformational transition[44].

A major finding of this study is the identification of direct lipid–protein interactions between the charge-relay networks of MFS transporters and the amine headgroup of the PE phospholipid. The synergistic power of MD and HDX-MS revealed a key role for this interaction in the conformational regulation of XylE and LacY. Indeed, while differential HDX-MS experiments indicated a shift toward the IF state in PE-based nanodiscs, MD simulations showed how the PE headgroup promotes this conformation by preventing the contacts required for the conformational transition. PE interacts with the intracellular ends of the so-called gating-helices (TM5-TM8 and TM2-TM11) and hinders the formation of an N-lobe and C-lobe interface (Fig. 4c). Even more striking is the motion of the PE phospholipid inside the transporter, interacting directly with residues E153, R84, and D337 of XylE (Fig. 4b, c and Supplementary Movie 1). Similar direct interactions between PE and residue E139 of LacY (Supplementary Fig. 10 and Supplementary Movie 2) were also

observed, in line with a previous computational study of LacY in different lipid environments[45].

These interactions shed light on the central role of these conserved charge-relay networks. Our study suggests that such networks are crucial not only in the stabilization of the OF state, but of the IF state as well. In this scenario, direct interactions between the PE headgroup and residues forming the networks stabilize the IF state. This hypothesis agrees with the findings of a previous study on the MFS transporter LmrP which used DEER spectroscopy to show that the conformational equilibrium was shifted toward the IF state in DOPE-based nanodiscs[19]. Furthermore, a number of studies have shown significant functional differences for the transporters LacY[46,47], MelB[48] from *E. coli*, LmrP[49] from *Lactoccocus lactis*, and the branched-chain amino acid transport system of *Streptococcus cremoris*[50] reconstituted in PE- and PC-based liposomes. These studies all indicated that the presence of PE in the lipid bilayer improves transport efficacy. We speculate that these functional differences arise partly from a shift of the conformational equilibrium caused by the lack of PE. The absence of direct interactions between PE and the transporter would give rise to a conformational imbalance limiting transport turnover. However, long-range effects such as lateral pressure and membrane curvature are also present in liposomes and in vivo, and likely to affect function as well.

Our findings allow us to reconsider the current alternating-access model for MFS transporters. A comprehensive description of a transport cycle rests on the ability to identify specific pathways along a free energy landscape that allow interconversion

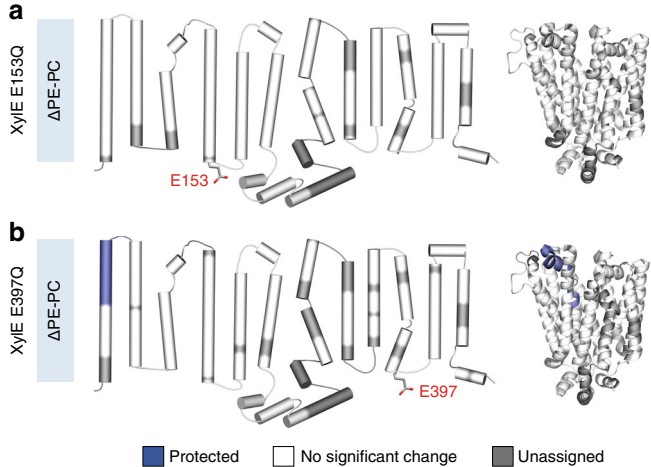

**Fig. 6** Disruption of conserved charge networks abolishes or decreases lipid-induced conformational shift. ΔHDX of XylE in PE:PG:CL nanodiscs (native-like) minus PC:PG:CL (control) nanodiscs mapped on the PDB structure and the topological representation of XylE for (**a**) XylE E153Q and (**b**) XylE E397Q. All measurements were performed in triplicates

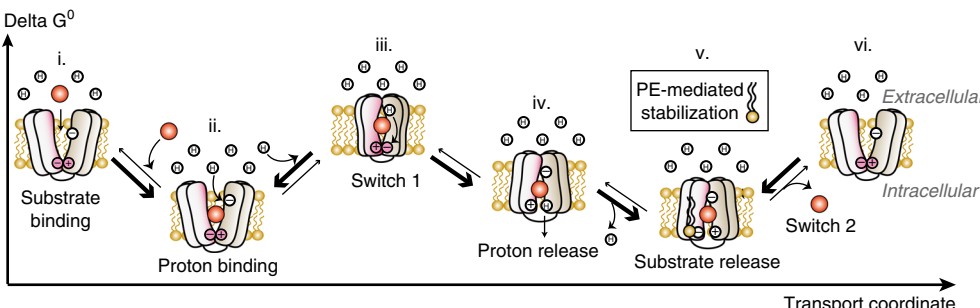

**Fig. 7** A lipid-mediated mechanism of transport. Model of secondary transport based on the interaction between charge-relay networks of XylE and PE phospholipid (i) Substrate (orange) binds from the extracellular side and stabilizes the OF conformation. (ii) Proton binding triggers closure of the extracellular side. (iii) The fully loaded transporter transitions towards an occluded conformational intermediate. Proton translocates through the transporter to the conserved charge-relay networks (pink) on the intracellular side (iv) Disruption of the charge networks opens the intracellular side. Spontaneous deprotonation upon exposure to the slightly basic cytosol leads to intracellular proton release. (v) Direct interactions between PE and the transporter stabilize the inward-facing conformation, thus facilitating substrate release and completing the transport cycle. Reformation of the cytoplasmic charge-relay networks leads to the subsequent conformational switch. (vi) is equivalent to (i)

between IF and OF conformations. Alongside other studies, this work demonstrates that the depiction of such landscapes must account for the role of the surrounding lipid environment. We propose an updated transport mechanism that includes direct protein–lipid interactions, exemplified by the proton-coupled symporter XylE and its interaction with PE (Fig. 7). Substrate (Fig. 7 – i) and proton (Fig. 7 – ii) binding from the periplasm to the OF conformation initiates the conformational transition towards an occluded state (Fig. 7 – iii). In the case of the symporter XylE, our work and other studies[51] suggest that proton binding at residue 27 will trigger the extracellular closing. The proton would then travel from residue 27 to one of the conserved acidic residues of the cytoplasmic charge networks and disrupt the stabilizing charge networks (Fig. 7 – iv). The disruption of these networks will trigger the opening of the protein on the intracellular side. The exposure to the slightly basic intracellular pH will lead to spontaneous deprotonation and intracellular proton release. Direct interactions between PE lipids and the charge networks will stabilize IF conformation, thus facilitating substrate release into the cytosol (Fig. 7 – v). The apo transporter can transition back to an OF conformation and start a new transport cycle (Fig. 7 – vi). In secondary active transport, the

substrate release and subsequent conformational switch of the transporter after a complete cycle is the rate-limiting step[52,53]. Our model suggests that a balance between formation of a charge network on the intracellular side (which favors the OF state) and direct interactions of the charged residues with PE (which favor the IF state) defines the energy barrier of this conformational transition, thus setting up the optimal transport rate.

It is interesting to note that the residues forming the cytoplasmic networks observed on XylE are also present on the mammalian homologs GLUTs1-5, which function as uniporters. Comparison of the crystal structures of GLUT3[54] and GLUT5[55] (captured in outward-open and outward-occluded conformations, respectively) with the outward-occluded conformation of XylE indicate that they share an identical structural arrangement (Supplementary Fig. 2). Mutations in the corresponding charge networks of GLUT1 have been associated with glucose deficiency syndrome[56]. These findings underline the importance of these networks even in transporters that are not proton-coupled and support our hypothesis that other molecular mechanisms, such as direct lipid–protein interactions, play a role in the conformational regulation underpinning function.

However, the conservation of these networks in transporters from different species does not coincide with conservation of the lipid bilayer composition. While PE lipids are the main component of the inner bacterial membrane of *E. coli*, their distribution is very diverse in other forms of life. In mammals, the chemical diversity of membrane lipids is much more complex and PE-based phospholipids are a minor species in most mammalian membranes[57]. It is possible that other types of protein–lipid interactions play a similar role as conformational regulators. For example, a recent study has shown the requirement of anionic phospholipids for the activity of GLUT 3 and 4 and demonstrated that this functional regulation happens through direct lipid–protein interactions[58]. Another secondary transporter, the dopamine neurotransmitter DAT is sensitive to the presence of the lipid PIP2 and MD simulations suggest that this functional regulation occurs through direct lipid–protein interactions[20]. We speculate that conformational regulation by specific lipid–protein interactions is a widespread mechanism followed by many other transporters.

Our approach that combines MD simulations with HDX-MS in tuneable nanodiscs provides an important new window into the relationship between structural and sequence conservation of transporters and membrane diversity across different animal kingdoms. This work widens our understanding of the molecular mechanisms governing transporter dynamics and provides a framework to study similar systems, such as the clinically relevant SLC transporters[59].

## Methods

**Materials.** Lipids were purchased from Avanti Polar Lipids Inc and detergents from Anatrace. Unless otherwise stated, all other chemicals were purchased from Sigma-Aldrich.

**Design and construction of the mutants.** The mutations were introduced in the C-terminally histidine-tagged proteins by site-directed mutagenesis using a Quik-Change Lightning kit (Stratagene). Primers are listed in Supplementary Table 1. After transformation, plasmid DNA was extracted using Qiaprep miniprep kit and verified by sequencing (Genewiz, UK). The plasmid containing the desired mutation was transformed in *E. coli* BL21-AI cells (Invitrogen) for overexpression.

**Protein expression and purification.** LacY, GlpT, and XylE were overexpressed in *E. coli* BL21-AI (Invitrogen) and purified using the same protocol as previously described[14] with the following modifications. Briefly, BL21-AI cells were transformed with the lacy, xyle, or glpt gene – with or without the chosen mutations – cloned in the kanamycin-resistant pET28 plasmid (Novagen) with a C-terminal 10-histidine tag, grown in LB media at 37 °C 250 r.p.m. to an OD$_{600}$ of 0.8. Expression was induced with 1 mM IPTG and 0.1% (w/v) arabinose and growth continued until saturation. The cells were harvested, passed through a microfluidiser (Constant Systems), and the membranes isolated by centrifugation for 30 min at 100,000 × g. Membrane proteins were solubilised for 2 h in 50 mM sodium phosphate, 100 mM NaCl, 10% (v/v) glycerol, 20 mM imidazole, 2 mM β-mercaptoethanol, pH 7.4 with 2% (w/v) β-DDM (Anatrace). In the case of GlpT, sodium phosphate was replaced in all buffers by 50 mM Tris-HCl to avoid phosphate binding. Solubilized protein was bound to a 1 mL Histrap column (GE Healthcare) and washed with buffer containing 75 mM imidazole and 0.05% (w/v) β-DDM. Purified protein was eluted with 500 mM imidazole and exchanged into a final buffer of 50 mM sodium phosphate for XylE and LacY, or 50 mM Tris-HCl for GlpT, 100 mM NaCl, 10% (v/v) glycerol, 1 mM β-mercaptoethanol, 0.02% β-DDM pH 7.4 by gel filtration chromatography. The samples were either flash frozen and kept at −80 °C until use or concentrated on Amicon 50 K concentrators and used directly for HDX-MS experiments. It is worth noting that the mutant GlpT E153Q was designed based on sequence conservation but was not kept for further analysis because of poor stability and aggregation.

**MSP1E3D1 production and purification.** Membrane scaffold protein (MSP1D1E3) was expressed and purified as previously described[19], with the following modifications. Briefly, *E. coli* BL21(DE3) (New England Biolabs) cells containing the MSP1D1E3 gene in pET-28a(+) were plated on LB-agar plates supplemented with kanamycin (30 μg mL$^{-1}$). A single colony was used to inoculate 30 mL of LB supplemented with 30 μg mL$^{-1}$ of kanamycin. A dense overnight culture of 30 mL was used to inoculate a secondary culture in 1 L Terrific broth supplemented with 30 μg mL$^{-1}$ kanamycin. Cultures were grown at 37 °C with shaking to an OD600 of ~2.2–2.5, and then expression of MSP1D1E3

was induced by addition of 1 mM IPTG. Cultures were further grown for 4 h at 37 °C, and cells were harvested by centrifugation. Cell pellets were resuspended in 30 mL of lysis buffer (20 mM sodium phosphate, 1% Triton X-100, pH 7.4), including one-third of a complete EDTA-free protease-inhibitor-cocktail tablet (Pierce), 10 μg mL$^{-1}$ benzonase I, and were lysed by two passes at 15,000 psi in a high-pressure homogenizer. The lysate was centrifuged at 30,000 × g for 30 min, and the supernatant was mixed with 3 mL of Ni–NTA resin equilibrated with lysis buffer. The slurry was transferred to a column, and the flow-through was discarded. The resin was washed with four bed volumes of buffer A (40 mM Tris-HCl and 0.3 M NaCl, pH 8.0) containing 1% Triton X-100, four bed volumes of buffer A containing 50 mM sodium cholate, four bed volumes of buffer A containing 20 mM imidazole and four bed volumes of buffer A containing 50 mM imidazole. The bound protein was eluted stepwise with buffer A containing 300 mM imidazole. The eluted MSP1D1E3 was passed over a desalting column into MSP buffer (50 mM Tris-HCl, 0.1 M NaCl, and 0.5 mM EDTA, pH 7.5), and the concentration was determined on the basis of the absorbance at 280 nm (extinction coefficient = 29,910 M$^{-1}$ cm$^{-1}$). The protein was concentrated to ~15 mg mL$^{-1}$ on a 10 K MWCO concentrator (Amicon). The purity was assessed by SDS–PAGE and Coomassie staining. The 6-his tag was removed by overnight incubation at 4 °C with TEV protease (80 μg of TEV to cleave 1 mg of MSP). The non his-tagged MSP were collected by reverse IMAC, and concentrated again to ~15 mg mL$^{-1}$.

**Lipid preparation for nanodiscs.** Lipids DOPE, DOPE(Me)$_1$, DOPC, DOPG, 18:1 CL dissolved in chloroform (Avanti Polar Lipids) were combined in a 7:2:1 ratio to reach a final quantity of 100 mg, dried under nitrogen flow and then desiccated overnight under vacuum. The lipid films were hydrated with MSP buffer to reach a final concentration of 40 mg mL$^{-1}$. β-DDM was added to the mixture to a final concentration of 7.5% (w/v). The lipids were further homogenized by four cycles of freeze-thawing, divided into 100 μL aliquots and stored at −80 °C. The choice of the chain length was motivated by the low T$_m$ temperature of dioleoyl chains, which facilitates the reconstitution of the membrane proteins at 4 °C.

**Reconstitution of MFS transporters into nanodiscs.** For reconstitution into nanodiscs, WT or mutants in β-DDM micelles were mixed with the appropriate DOPE or DOPC-based lipid mixture, MSP1D1E3, and β-DDM in the following molar ratios: lipid/MSP1D1E3, 60:1; MSP1D1E3/XylE, 8:1; and β-DDM/lipid, 3:1. Samples were rocked at room temperature for 30 min then incubated overnight at 4 °C with rocking. Biobeads SM-2 (700 mg mL$^{-1}$) (Bio-Rad) were added to the mixture and incubated for 2 h at 4 °C, then 1 h at RT. The removal of the Biobeads was achieved by low-speed centrifugation of the nanodisc assembly in tubes that had been perforated with a needle to form a tiny hole. Full nanodiscs were separated from empty ones on Ni-NTA high-affinity beads (500 μL of beads for 1 mg of reconstituted XylE), eluted with nanodiscs purification buffer (sodium phosphate 50 mM, 100 mM NaCl, 10% glycerol) supplemented with 300 mM imidazole. The full disks were further purified by size-exclusion chromatography on a Superdex200 column (GE) using nanodiscs purification buffer, then were concentrated to 100 μL final volume with Amicon Ultra-50K centrifugal filter units at a speed not exceeding 2000 g. Full nanodiscs were then characterized with SDS–PAGE to assess reconstitution and DLS to ensure homogenous size of the nanodiscs. All DLS measurements were carried out on a Zetasizer Nano ZS (Malvern).

**Hydrogen–deuterium exchange mass spectrometry.** Hydrogen–deuterium exchange mass spectrometry (HDX-MS) experiments were performed on a Synapt G2Si HDMS coupled to an Acquity UPLC M-Class system with HDX and automation (Waters Corporation, Manchester, UK). Membrane proteins samples in detergent micelles (see the next section for nanodiscs samples) were prepared at a concentration of 25 to 70 μM using Vivaspin concentrators with a 50,000 MWCO cutoff. Isotope labeling was initiated by diluting 5 μl of each protein sample into 95 μL of buffer L (10 mM potassium phosphate in D$_2$O pH 6.6). The protein was incubated for 30 s, 5 min, and 30 min to capture short, medium and long exchange times and then quenched in ice cold buffer Q (100 mM potassium phosphate, brought to pH 2.3 with formic acid) before being digested online with a Waters Enzymate BEH pepsin column at 20 °C. The same procedure was used for undeuterated controls, with the labeling buffer being replaced by buffer E (10 mM potassium phosphate in H$_2$O pH 7.0). The peptides were trapped on a Waters BEH C18 VanGuard pre-column for 3 min at a flow rate of 200 μL min in buffer A (0.1 % formic acid ~ pH 2.5) before being applied to a Waters BEH C-18 analytical column. Peptides were eluted with a linear gradient of buffer B (8–40% gradient of 0.1 % formic acid in acetonitrile) at a flow rate of 40 μL min$^{-1}$. All trapping and chromatography was performed at 0.5 °C to minimize back exchange. The electrospray ionization source was operated in the positive ion mode and ion mobility was enabled for the instrument. MSE data were acquired with a 20–30 V trap collision energy ramp for high-energy acquisition of product ions. Leucine Enkephalin (LeuEnk, Sigma) was used as a lock mass for mass accuracy correction and the mass spectrometry was calibrated with sodium iodide. The online Enzymate pepsin column was washed with pepsin wash (1.5 M Gu-HCl, 4 % MeOH, 0.8% formic acid) recommended by the manufacturer and a blank run using the pepsin wash was performed between each sample to prevent significant peptide carry-over

from the pepsin column. Optimized peptide identification and peptide coverage for all samples was performed from undeuterated controls (five replicates). All deuterium time points were performed in triplicate.

**Data evaluation and statistical analysis**. Sequence identification was made from MSE data from the undeuterated samples using ProteinLynx Global Server 2.5.1 (PLGS Waters Corp. Manchester, UK). The output peptides were filtered using DynamX (v. 3.0), using the following filtering parameters: minimum intensity of 1000, minimum and maximum peptide sequence length of 5 and 25, respectively, minimum MS/MS products of 2, minimum products per amino acid of 0.25, minimum score of 5, and a maximum MH+ error threshold of 15 p.p.m. Additionally, all the spectra were visually examined and only those with high signal to noise ratios were used for HDX-MS analysis. The amount of relative deuterium uptake for each peptide was determined using DynamX (v. 3.0) and are not corrected for back exchange since only relative differences were used for analysis and interpretation and there was no benefit from normalizing the data[60]. The relative fractional uptake (RFU) was calculated from $RFU_a = [Y_{a,t}/(MaxUptake_a \times D)]$, where Y is the deuterium uptake for peptide a at incubation time t, and D is the percentage of deuterium of the present in the sample after mixing the protein with the labeling solution. Confidence intervals for the summed ΔHDX over the three time points were then determined according to Houde et al[61]. Specifically, the significance of the summed difference was assessed by a t-test ($n = 3$; $P \le 0.01$, two-sided, unpaired) to evaluate the overall difference in uptake. All the peptides passing the test were mapped on the 3D structure and topological map of the protein of interest, using an in-house script which follows a binary color scheme, in which any positive ΔHDX (variable minus control) is red and any negative ΔHDX is blue. The code is available on GitHub code repository (https://github.com/andymlau/Deuteros).

**Preparation of nanodiscs samples for HDX-MS**. We found that the nanodisc concentration was a critical parameter to obtain a good sequence coverage and the HDX conditions were manually optimized for each nanodiscs sample. Practically, the purified and concentrated nanodiscs were tested for coverage by performing a reference only run on the mass spectrometer and the coverage was immediately assed using PLGS. The protein was concentrated using mini vivaspin concentrators (Amicon, cutoff 50 K MWCO) until a coverage of more than 85% was achieved. Then, the optimal sample scheme for HDX purposes was as follows: 5 μL of nanodiscs sample was diluted into 95 μL of nanodiscs purification buffer or with deuterated buffer nanodiscs purification buffer at 22 °C. The deuterated samples were left for 5 min, 30 min, and 150 min at 22 °C in a dry bath. All samples were then quenched with 100 μL of quench buffer (100 mM potassium phosphate, brought to pH 2.4 with formic acid, 0.2% DDM). The sample was vortexed for 3 s then placed immediately on ice for 1 min. Ten microliter of $ZrO_2$ beads (300 mg/mL equilibrated in quench buffer) was then added to remove lipids and the sample vortexed for 3 s, placed on ice for 30 s, vortexed for a further 3 s, and then placed on ice for a further 30 s. The samples were then filtered through pre-chilled 0.22-μm spin filtration devices (Corning Costar Spin-X) in a pre-chilled micro-centrifuge at 1000 xg for 30 s. Digested or undigested samples were then immediately flash frozen in liquid nitrogen and stored at −80 °C before analysis. Samples were rapidly defrosted and then injected into a Waters HDX nanoAcquity ultra-performance liquid chromatography (UPLC) system (Waters) using a pre-chilled syringe, as described in the previous section.

**Molecular dynamics (MD) simulations setup**. All the simulations were initiated from the IF-open state of XylE (PDB ID:4JA4)[10]. Missing loops and helices were modeled using superlooper[62] using the OF-open state of XylE (4GBY as the template). Assignment of the protonation state was done on the basis of pKa calculations performed using PROPKA3.1 at pH 7[63]. Accordingly, all the glutamate and aspartate except D27 were modeled in their default (unprotonated) form. Using the membrane replacement method in CHARMM-GUI[64], XylE was embedded in lipid bilayers of two compositions (DOPE-DOPG-CL (70:20:10) and DOPC-DOPG-CL (70:20:10)), closely mimicking the experimental conditions. System was solvated with TIP3P water molecules[65]. Thereafter, $Na^+$ and $Cl^-$ ions were added, and the system was neutralized with the ionic concentration set to 100 mM. The final system inclusive of the protein, lipids, water molecules, and ions comprised ~105 K atoms. Thereafter, the system was minimized for 5000 steps using conjugate-gradient algorithm and simulated for 5 ns at 310 K, with all the heavy atoms of the protein restrained to their crystallographic positions with a force constant of k = 5 kcal/mol/Å². In the next step, the system was further simulated for 5 ns with harmonic restraints (k = 5 kcal/mol/Å²) applied only to the parts of the protein that was modeled using superlooper. Finally, all the restrains were removed and the systems were simulated for 500 ns.

**MD simulation protocol**. The simulations were performed on with NAMD 2.12[66] utilizing CHARMM36 all-atom forcefields for proteins and lipids[67]. NPT ensemble with periodic boundary conditions was used for all simulations. Simulations were performed at 310 K using Langevin dynamics[68] with a damping constant of 0.5 ps⁻¹. Pressure was maintained at 1 atm using the Nosé–Hoover Langevin piston method[68]. The cutoff used for the short-range interactions were 12 Å with the

switching applied at 10 Å. We used particle mesh Ewald (PME) algorithm to calculate the long-range electrostatic forces[69]. Bonded, non-bonded, and PME calculations were performed at 2-, 2-, and 4-fs intervals, respectively.

## Data availability
Data supporting the findings of this paper are available from the corresponding author upon reasonable request. All the deuterium uptake plots and uptake datasets of the experiments presented in this work are available on figshare data repository. XylE data can be accessed using the following https://doi.org/10.6084/m9.figshare.7072988. LacY data can be accessed using the following https://doi.org/10.6084/m9.figshare.7073072. GlpT data can be accessed using the following https://doi.org/10.6084/m9.figshare.7073003. Furthermore, the mass spectrometry proteomics data have been deposited to the ProteomeXchange Consortium via the PRIDE[70] partner repository with the dataset identifier PXD011060.

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

## Acknowledgements

We thank Euan Pyle, Matthieu Masureel and Bernadette Byrne for critically reading the manuscript, Heather Findlay and Maxwell Allen-Benton for their precious help during the reviewing process and the Booth and Politis groups for helpful discussions. This work was supported by the Wellcome Trust (109854/Z/15/Z) and a King's Health Partners R&D Challenge Fund through the MRC (MC_PC_15031) to A.P, and ERC advanced grant 294342 to P.J.B. E.R. is funded by a BBSRC Future Leader Fellowship BB/N011201/1. The research is also supported in part by the National Institute of General Medical Sciences of the National Institutes of Health under awards U54-GM087519, P41-GM104601, and R01-GM123455 to E.T. We also acknowledge computing resources provided by Blue Waters at National Center for Supercomputing Applications, and Extreme Science and Engineering Discovery Environment (grant TG-MCA06N060 to E.T.).

## Author contributions

C.M, P.J.B, and A.P. conceived the study and designed the experiments. C.M. performed the mutagenesis, expression, purification, and reconstitution experiments. C.M. and A.B. carried out the HDX-MS experiments. M.S. and E.T. performed and analyzed the MD simulations. C.M. and A.M.L. analyzed the HDX-MS data. C.M. and E.R. optimized the HDX-MS protocol for nanodiscs. C.M. and A.P. wrote the paper with input from all authors.
