## [Peer Review File · Nature Communications]

Reviewers' comments:

Reviewer #1 (Remarks to the Author):

This manuscript uses HDX-MS on three proteins (XylE, LacY and GlpT) to probe the influence of conserved charged residues on protein conformations as well as how lipids influence XylE conformational dynamics. Complement to the experimental work are molecular simulations on XylE. Overall, this is a well written and timely manuscript that provides unlabeled detail on regions of the protein that observe structural changes in a semi-native lipid environment. My specific comments are listed below.

Comments:

1. Cartoon figures: In all the figures with cartoon images (Figure 1b is one example, but there are many others in the main and supporting materials), it would be nice to highlight the general location of protein mutated residues so the exposed/protected changes can structurally be related to the mutation.
2. LacY and lipid interaction: There is a paper by Andersson et al. (*Structure*. 20: p1893, 2012) that discovered interactions of the lipid PE headgroup with E139 but not D68. The authors should consider reading this work and how this might add to the lipid-protein interaction work with XylE presented in the current manuscript as well as the conserved charge residue work directly with LacY.
3. Nanodisc lipids: The authors state that nanodiscs are made with 7:2:1 molar ratio of PE(or PC), PG and CL. Is this the actual distribution in the nanodisc as this started with the lipid films? Also, DDM was present in these mixtures and might in partition into the nanodisc?
4. Figure 4 caption: the wording of "but no such thing is seen" is a bit informal and should be reworded.

Reviewer #2 (Remarks to the Author):

Martens and colleagues present a strong case for the role of lipids in the conformational changes in secondary sugar transporters. They combine HD-exchange MS with MD simulations to probe the role of lipids in the conformational dynamics of XylE and to lesser extent LacY and GlpT. The authors conclude that PE favors the inward conformation of the proteins, and on the basis of new structural data they present a new lipid-mediated mechanism of transport for XylE. The work is well done and I only have a few comments to be addressed.

1. The authors should relate their structural data to functional data, which can be done in proteoliposomes of the same lipid composition as used for the nanodiscs. A lot of functional work on the lipid dependence has been done with LacY (and to lesser extent if at all with XylE and GlpT). For instance, the Kaback and Dowhan laboratories have shown that lactose-proton symport is more sensitive to changes in the lipid composition of the membrane (and other factors) than for instance equilibrium exchange or counterflow. Relevant information can also be found in (*Mol Membr Biol*. 2014 Jun;31(4):120). The authors should incorporate this work into their paper and present similar measurements for the lipid compositions they study (70% PE versus 70% PC).
2. Can the authors rule out (experimentally) topological switching of part of the protein as shown for LacY in membranes devoid of PE (e.g. *Proc Natl Acad Sci U S A*. 2015 Nov 10;112(45):13874)?

3. What is the dissociation constant for galactoside binding of LacY in PE and PC nanodiscs?

4. The author show that 70% PE drives the proteins to the inward conformation. Is this a specific PE effect or is it due effects on the lateral pressure profile of the membrane. For other transporters, most notably an ABC transporter it has been shown both in lipid vesicles and in nanodiscs that the activity increases linearly with fraction of PE in the membrane (EMBO J. 2001 Dec 17;20(24):7022; J Biol Chem. 2013 Oct 11;288(41):29862). This indicates that some general membrane property is affecting the transporter rather than a specific lipid-protein interaction. The question is whether 70% PE is needed to favor the inward state or that perhaps a few percent is sufficient. The authors should try to dissect a specific PE effect (the specific lipid-protein interaction they suggest) from a general effect resulting from the large fraction of non-bilayer type of lipids they use in the 70% PE membranes.

5. What is the evidence that secondary transporters in the outward-facing conformation leak more substrate and / or protons (as suggested in the model on page 20 and figure 7) than proteins that are in the inward-facing conformation? Do LacY-containing vesicles with PC rather leak more protons than those containing PE?

6. Finally, the manuscript should be checked for typos.

Reviewer #3 (Remarks to the Author):

This is a well written, high quality manuscript that addresses the mechanism of transmembrane transport for three model proteins for which X-ray structural data are available. HDX-MS experiments and MD simulations are used as complementary tools to decipher the involvement of specific membrane lipids and AA clusters in conformational switching.

A couple of points should nonetheless be addressed, prior to accepting this work for publication [none of these issues are very severe – all will be easy to fix]

*** Main Comments:

Introduction and/or abstract: For non-experts, please define what exactly is a “secondary” transporter. Also, are we discussing symport or antiport? Involvement of a proton gradient is neglected for the longest time within this manuscript, until very late (Figure 7).

Fig 1 caption should be improved: 1. “dHDX (red – more deuteration for G-W compared to WT) or negative dHDX (blue – less deuteration for G-W compared to WT)” 2. Delete the fictitious D uptake curves (top right). Let the experimental data speak for themselves. 3. Figure and caption: simplify by not attempting to explain the experimental workflow. 4. Avoid using “accessibility” when discussing HDX data, instead the text should focus on H-bond stability as commonly done in the HDX literature. 5. Top left: G-W and WT are both shown in the OF conformation; to casual readers it will not be clear how the protein conformation changes after mutation.

The authors are probably correct that the G-W mutation stabilizes the OF mutations. An alternative interpretation of Fig. 1b could be that the bulky Trp side chains causes disorder in its direct vicinity, thereby triggering the “red” regions in Fig. 1b. The authors may wish to address this possibility.

Similarly, can the authors rule out that the mutations in the intracellular acidic sites (Section 2) cause local conformational destabilization (“red”), without shifts of the IF/OF equilibrium?

Figure 7 and associated text: The transport cycle is unclear. (1) The proton is missing. (2) What is the evidence that PE binds *inside* the transport channel? (3) Clearly indicate intra- and extracellular side.

Methods: "and D is the percentage of deuterium in the final labeling solution" Please be more specific. Is this the 30 min time point? Also: The authors probably have a good reason why they did not use the standard approach of normalizing the HDX data to fully exchanged controls – this should be briefly explained.

*** Additional Minor Comments:

Methods: "TIP3P model was used for water." This is stated twice

p. 13 typo: We performed two setS ...

Methods: "Simulations were performed at 310 K using Langevin dynamics⁷⁹ with a damping constant of 0.5 ps⁻¹." I don't think this is a correct statement. Langevin dynamics usually refer to solvent-free simulations, whereas here the authors used explicit solvation by TIP3P water.

Introduction: "HDX reports the exchange of labile amide protons" change to "...HYDROGENS"

To streamline the text, consistently use "cytoplasmic side" or "intracellular side" but not a mix of both.

p. 20: "primarily found in the IF conformation, limiting substrate and/or proton leakage" This implies that OF would result in substrate and/or proton leakage ... but probably that's not what the authors are trying to say here?

Figure 5c: Why is the normalized count shown on a scale that goes up to 2.0? What do these units mean?

Introduction: Consider rewriting this sentence: "it enables conformational characterization of unlabelled, unmodified proteins" because the proteins detected by MS have been labeled (and hence modified). [it will be clear to many readers what the authors mean here, but still the wording is a bit misleading]

Point-by-point response

Reviewer #1 (Remarks to the Author):

This manuscript uses HDX-MS on three proteins (XylE, LacY and GlpT) to probe the influence of conserved charged residues on protein conformations as well as how lipids influence XylE conformational dynamics. Complement to the experimental work are molecular simulations on XylE. Overall, this is a well written and timely manuscript that provides unlabeled detail on regions of the protein that observe structural changes in a semi-native lipid environment. My specific comments are listed below.

Comments:

1. Cartoon figures: In all the figures with cartoon images (Figure 1b is one example, but there are many others in the main and supporting materials), it would be nice to highlight the general location of protein mutated residues so the exposed/protected changes can structurally be related to the mutation.

We have now modified the figures to include the location of the mutated residue on the topological maps of each protein. We do not show the location of the mutation on the 3D structure because in many cases the residue was not visible, so we avoided it altogether for consistency.

2. LacY and lipid interaction: There is a paper by Andersson et al. (Structure. 20: p1893, 2012) that discovered interactions of the lipid PE headgroup with E139 but not D68. The authors should consider reading this work and how this might add to the lipid-protein interaction work with XylE presented in the current manuscript as well as the conserved charge residue work directly with LacY.

We thank the reviewer for bringing our attention to this very relevant paper, which we have now included in our discussion (p.19). This comment, alongside reviewer's 2 comments, prompted us to run simulations of LacY in DOPE:DOPG:CL and DOPC:DOPG:CL bilayers. The simulations showed clear interactions between E139 and the PE headgroup. On par with our findings on XylE and the findings of the Anderson et al. paper mentioned above, no such interaction was observed in DOPC:DOPG:CL bilayers. These new results are presented in Figure 1 below.

Figure 1: MD simulations predict specific interactions between the PE headgroup and acidic residue E139 of LacY (a) Representative snapshot of the entrance of the PE phospholipid into the cytoplasmic space of LacY. Steric hindrance prevents contacts of TM5 and TM8. (b) Close-up of the interaction between the amine headgroup and the acidic residue E139. (c) Location of the phosphorus atoms of the PE/PC lipid molecule in 500 ns trajectories in PE:PG:CL bilayers (left) and PC:PG:CL bilayers (right). The spheres are colour coded: red at $t=0$ ns and blue at $t = 500$ ns. PE lipids wedge between the helices but no lipid-protein interaction is observed in PC bilayers.

To gain insight into the role of these lipids in LacY conformational dynamics, we carried out additional HDX-MS experiments of LacY in nanodiscs of the same lipid compositions as the ones used for nanodiscs containing Xyle. We found a similar result: the presence of PE lipids favours an opening on the cytoplasmic side (Figures 2 and 3).

Figure 2: Δ HDX of LacY WT in PE vs PC nanodiscs suggests that PE promotes the IF conformation. Δ HDX of WT LacY in DOPE-PG-CL nanodiscs (native-like) minus DOPC-PG-CL (control) nanodiscs mapped on the PDB structure (PDB ID 2V8N) and a topological representation. (b) Representative deuterium uptake plots for peptide 217-224, (c) 279-293, (d) 308-313 in DOPC-PG-CL nanodiscs (green) and DOPE-PG-CL nanodiscs (black). All measurements were performed in triplicates.

Figure 3: Woods plot of differences in relative deuterium uptake (Δ HDX) of LacY. The length of the lines represents the length of the peptide. CI1 (dotted lines) and CI2 (dashed lines) represent the 99% and 98% confidence interval respectively. The approximate location of the peptides is reported on the topological map of LacY.

We believe these data supports our hypothesis and strengthens our conclusions in line with our previous experiments with XylE, and should be added in the manuscript, if the editor agrees. These data are included in the manuscript as supplementary Fig. 10 for the MD simulations and supplementary Fig. 9 for the HDX-MS experiment.

3. Nanodisc lipids: The authors state that nanodiscs are made with 7:2:1 molar ratio of PE(or PC), PG and CL. Is this the actual distribution in the nanodisc as this started with the lipid films? Also, DDM was present in these mixtures and might in partition into the nanodisc?

We thank the reviewer for these pertinent questions. To demonstrate that the lipid composition of the lipids films used for reconstitution is identical to the actual composition of the nanodisc, we performed Thin Layer Chromatography (TLC) experiments (Entezami, Venables et al. 1987). We extracted the lipids from the XylE nanodiscs following established protocols (Bligh and Dyer 1959) and compared it to the starting lipid mix. As it can be seen on the TLC plates below (Figure 4), the composition and the ratio appear to be similar in both conditions. We quantified the spot intensities using the free software ImageJ (Schneider, Rasband et al. 2012) and obtained values in the range of the 7:2:1 ratio used for the films. The actual numbers differ from this initial ratio but we attribute this to the low resolution of the method. However, it is clear that the ratios of the lipids films and the lipids extracted from the discs are similar, which answers the reviewer's question: the nanodisc reconstitution procedure does not select specific lipid species. A number of other studies characterizing heterogeneous nanodiscs came to the same conclusion using either mass spectrometry (Hoi, Robinson et al. 2016) or TLC (Wadsater, Maric et al. 2013, Dijkman and Watts 2015).

Fig. 4: Thin Layer Chromatography of phospholipids starting mixes and nanodiscs. Comparison between lanes 4 and 5 on the left plate (DOPE:PG:CL) and lanes 9 and 10 on the right plate (DOPC:DOPG:CL) shows that the same lipid species are present in both samples.

Regarding the reviewer's concern about the presence of detergent, we performed native mass spectrometry analysis of the LacY nanodiscs preparation. For the sake of comparison, we also recorded a spectrum of LacY in DDM micelles. The presence of DDM is characterized by a peak at 533 m/z (mass of DDM + one Na⁺ ion). The native mass spectra are presented in Figure 5. It is clear from the observation of the low m/z region there is no detergent present in the nanodiscs samples. This data has now been added in the manuscript as supplementary Fig. 8.

Figure 5: Mass spectra of (a) LacY in PC nanodiscs, (b) LacY in PE nanodiscs, (c) LacY in detergent micelles. The detergent and free lipids are shown in the low m/z region on the left and the protein with bound lipids is shown in the high m/z region on the right. The position of the DDM peak is indicated by a red rectangle.

4. Figure 4 caption: the wording of “but no such thing is seen” is a bit informal and should be reworded.

This has now been changed to “no lipid-protein interaction is observed in PC bilayers”.

Reviewer #2 (Remarks to the Author):

Martens and colleagues present a strong case for the role of lipids in the conformational changes in secondary sugar transporters. They combine HD-exchange MS with MD simulations to probe the role of lipids in the conformational dynamics of XylE and to lesser extent LacY and GlpT. The authors conclude that PE favors the inward conformation of the proteins, and on the basis of new structural data they present a new lipid-mediated mechanism of transport for XylE. The work is well done and I only have a few comments to be addressed.

1. The authors should relate their structural data to functional data, which can be done in proteoliposomes of the same lipid composition as used for the nanodiscs. A lot of functional work on the lipid dependence has been done with LacY (and to lesser extent if at all with XylE and GlpT). For instance, the Kaback and Dowhan laboratories have shown that lactose-proton symport is more sensitive to changes in the lipid composition of the membrane (and other factors) than for instance equilibrium exchange or counterflow. Relevant information can also be found in (Mol Membr Biol. 2014 Jun;31(4):120). The authors should incorporate this work into their paper and present similar measurements for the lipid compositions they study (70% PE versus 70% PC).

We agree with the reviewer's comment that structural data should be linked with functional data. In response to the reviewer's comment, we have extended our structural characterization of LacY to study how the lipid environment modulates its conformational transition. Since LacY is the most extensively studied secondary transporter, we reasoned that it would be more sensible to provide new structural data and interpret it in the context of the widely available functional data. Indeed, no active transport assay is available for XylE and our initial attempts to develop a novel assay clearly indicated that substantial work would be required, which would go beyond the scope of this revision process.

Following the protocol established for XylE, we reconstituted the transporter LacY in PE:PG:CL and PC:PG:CL nanodiscs and performed HDX-MS measurements. We also extended our molecular dynamics simulations to LacY in lipid bilayers of similar compositions. The results – presented in figs.1 and 2 (see response to reviewer 1) – are similar to the ones observed for XylE. The presence of PE seems to favor the inward-facing conformation, and a direct interaction between PE and a conserved charged residue (here E139) is predicted to play a role in such conformational preference.

With this new data in hand, we reviewed the available literature on LacY lipid dependence. Multiple studies show that PE is required for active transport (Chen and Wilson 1984, Vitrac, Bogdanov et al. 2013, Findlay and Booth 2017), and that replacing it by PC does not affect either topology or facilitative diffusion but abolishes coupled transport. This finding supports the transport model presented in the discussion: the stabilization of the inward-facing conformation by direct lipid-protein interactions facilitates substrate release into the cytosol, which is the rate-limiting step of LacY active transport cycle (Kaback 2015). This literature review also revealed that

the exact lipid composition we used for our nanodiscs reconstitution has never been used in *in vitro* transport assays of LacY. Following a protocol established in the Booth group (Findlay and Booth 2017), we performed active transport assay of LacY reconstituted in proteoliposomes of the same lipid composition as the ones used for our HDX experiments (Figure 6 below), as requested by the reviewer. LacY was reconstituted into 200nm lipid vesicles with the enzyme β -galactosidase encapsulated. The substrate o-NPG was added externally. We then generated a pH gradient by diluting the proteoliposomes in a slightly acidic buffer (ext pH6.5/ int pH7.4). Transported o-NPG was digested to produce nitrophenol, which was monitored by the increase in absorbance at 410 nm. Empty proteoliposomes and proteoliposomes containing the secondary transporter GlpT were used as negative controls. As can be seen on the figure below, our results are in line with the literature: active import of LacY substrate is observed only in liposomes containing PE. No transport is observed in PC liposomes.

Figure 6: LacY transport activity in vesicles of different lipid composition. (a) Schematic of the transport assay. (b) Active transport of NPG by LacY in PE proteoliposomes. (c) No transport of NPG is observed in PC proteoliposomes.

We also agree with the reviewer that previous functional characterization of the transporters presented in the study should be acknowledged. We have now updated our discussion by referencing previous work that studied the role of lipids on the transport function of LacY (p.20).

2. Can the authors rule out (experimentally) topological switching of part of the protein as shown for LacY in membranes devoid of PE (e.g. Proc Natl Acad Sci U S A. 2015 Nov 10;112(45):13874)?

HDX is actually a method of choice for assessing topology. It is clear from our numerous datasets that the regions of the transmembrane helices that are embedded in the detergent micelle or the lipid nanodiscs do not undergo any HD exchange while the regions exposed to solvent all take up deuterium. If a topological switching occurs, the pattern of uptake will be drastically different.

Figure 7 below reports the relative deuterium uptake of LacY at the longest time point (2h 30min) for each peptide, in DOPC vs DOPE nanodiscs. The relative uptake values

at the maximum deuteration time per peptide are similar for both lipid compositions, which indicate that there is no change in topology. LacY topological switching has been shown to involve helix 7. This helix is highlighted on the figure and no obvious change in uptake occurs in that region.

Figure 7: Coverage map of LacY in PC nanodiscs (top panel) and PE nanodiscs (lower panel). The relative deuterium uptake is mapped on each peptide and color-coded on a gradient scale from green to red.

It is worth noting that according to different studies (Dowhan and Bogdanov 2011, Findlay and Booth 2017), a topology switch occurs in membranes lacking PE, but the

native topology is restored when PE is substituted for PC. In this regard, our findings are consistent with previous work, including the paper referenced by the reviewer. It appears from these studies that the topological switching observed in LacY is caused by an increase of anionic charges in the membrane (eg. increase in the amount of PG), and not by changes in the bilayer curvature (eg. replacing PE by PC).

3. What is the dissociation constant for galactoside binding of LacY in PE and PC nanodiscs?

To answer this question we performed ITC measurements of LacY reconstituted in nanodiscs of DOPE:DOPG:CL and DOPC:DOPG:CL. The data obtained was not conclusive and suggested unspecific binding since no saturation is observed (see Figure 8 below).

Figure 8: ITC measurements of LacY reconstituted in (a) nanodiscs of DOPE:DOPG:CL, (b) nanodiscs of DOPC:DOPG:CL and (c) DDM detergent micelles.

We presume that specific binding can't be detected because the presence of the scaffold protein belt (MSP) interferes with the signal. Furthermore, LacY affinity for its substrates is relatively low - in the μM range. Here we used 2-NPG, the substrate reported to have the highest affinity according to (Nie, Smirnova et al. 2006). It is worth noting that the nanodisc reconstitution procedure requires a lot of material (lipids, protein and scaffold protein) and ITC measurements uses up quite a lot of sample, which means that we couldn't test many conditions to find the perfect substrate to sample ratio. Furthermore, once reconstituted, it is impossible to measure the actual quantity of LacY because of the scaffold protein interference. We hope that the reviewer appreciates our effort in trying to answer his/her question but we believe that at this stage this information is not required to support our mechanistical model.

4. The author shows that 70% PE drives the proteins to the inward conformation. Is this a specific PE effect or is it due effects on the lateral pressure profile of

the membrane. For other transporters, most notably an ABC transporter it has been shown both in lipid vesicles and in nanodiscs that the activity increases linearly with fraction of PE in the membrane (EMBO J. 2001 Dec 17;20(24):7022; J Biol Chem. 2013 Oct 11;288(41):29862). This indicates that some general membrane property is affecting the transporter rather than a specific lipid-protein interaction. The question is whether 70% PE is needed to favor the inward state or that perhaps a few percent is sufficient. The authors should try to dissect a specific PE effect (the specific lipid-protein interaction they suggest) from a general effect resulting from the large fraction of non-bilayer type of lipids they use in the 70% PE membranes.

We thank the reviewer for pointing out these interesting studies that had escaped our attention. To answer the question, we reconstituted XylE in nanodiscs composed of DOPE with one methyl group on the amine headgroup (DOPE(Me)₁:DOPG:CL) and compared it with the DOPC:DOPG:CL lipid nanodiscs. Indeed, according to (Hamai, Yang et al. 2006), the curvature properties of DOPE decreases linearly with the addition of each methyl group on the amine. Hence, the curvature properties of the DOPE(Me)₁ nanodiscs should be comparatively closer to the DOPE than the DOPC nanodiscs. And if the observed shift in the conformational equilibrium is caused by the difference in curvature properties, such shift should still be observed. However, our HDX experiments show that there is no difference in uptake between the DOPC and DOPE(Me)₁ nanodiscs, which indicates that their conformational landscape is identical. This data is presented on Figure 9 and added in the manuscript as supplementary Fig. 11.

Figure 9. HDX-MS of XylE WT in nanodiscs of lipids with different intrinsic curvature. Δ HDX of WT LacY in DOPE(Me)₁-PG-CL nanodiscs (high curvature) minus DOPC-PG-CL (low curvature) nanodiscs mapped on the PDB structure (4GBY) and the topological representation. (b) Woods plot.

This data, alongside our mutagenesis experiments and MD simulations, converges toward the notion that a direct interaction is likely to be at play, at least in nanodisc. We do specify in the discussion that our experimental system doesn't represent an actual membrane and that additional bulk effects have to be considered *in vivo*.

5. What is the evidence that secondary transporters in the outward-facing conformation leak more substrate and / or protons (as suggested in the model

on page 20 and figure 7) than proteins that are in the inward-facing conformation? Do LacY-containing vesicles with PC rather leak more protons than those containing PE?

It appears from this comment and the third reviewer comments that our model was confusing. As such, we would like to clarify it here.

Multiple lines of evidence support the idea that LacY most stable conformation is the inward-open conformation. EPR studies have shown that the apo transporter is mostly in the inward-open conformation (Smirnova, Kasho et al. 2007) and it has so far been impossible to crystallize WT LacY in the outward-facing conformation. For these reasons, we are inclined to think that the resting state is the inward-facing state. We do find it unlikely to have an outward-open resting state as it would expose the transporter to the extracellular protons, which might reach the cytoplasmic charge networks and thereby open the cytoplasmic side, hence creating a futile cycle. However, there is no evidence supporting this scenario and it might well be that the extracellular protons do not reach the charge networks when the transporter is outward-open. We agree that the statement “In its resting state, the transporter is primarily found in the IF conformation, limiting substrate and/or proton leakage the inward-facing conformation limits substrate or proton leakage” was not supported by evidence and we have now removed it.

However, it is well established that both substrate and proton have to bind from the extracellular side, which implies that these binding events happen when the protein is in an outward-open conformation. It is also established that once the transporter is loaded with both ligands it will undergo a conformational transition. We have now rewritten the paragraph to start our model from this loading step (*see Discussion p.20 and 21*). We propose that the lipid-mediated stabilization of the inward-open conformation facilitates substrate release in the cytosol, which is the rate-limiting step of an active transport cycle. Such hypothesis reconciles our structural findings with the current functional data: the presence of PE is required to allow substrate release and subsequent conformational transition, and thus supports active transport. The model figure has been modified as well (p 21 and Fig. 7 in the manuscript)

6. Finally, the manuscript should be checked for typos.

We have done our best to fix the typos.

Reviewer #3 (Remarks to the Author):

This is a well written, high quality manuscript that addresses the mechanism of transmembrane transport for three model proteins for which X-ray structural data are available. HDX-MS experiments and MD simulations are used as complementary tools to decipher the involvement of specific membrane lipids and AA clusters in conformational switching. A couple of points should nonetheless be addressed, prior to accepting this work for publication [none of these issues are very severe – all will be easy to fix].

***** Main Comments:**

Introduction and/or abstract: For non-experts, please define what exactly is a “secondary” transporter. Also, are we discussing symport or antiport? Involvement of a proton gradient is neglected for the longest time within this manuscript, until very late (Figure 7).

We have now modified our introductory sentence to clarify this aspect. The first sentence now reads as “Secondary membrane transporters play crucial roles in maintaining adequate conditions for life by catalyzing uphill transport of biomolecules through the biological membrane, using energy stored in transmembrane ions gradients”.

This study does not differentiate antiporters from symporters as our work and others (Law, Almquist et al. 2008, Quistgaard, Low et al. 2016) strongly suggest that the formation/disruption of charge relay networks is a widespread mechanism, not limited to a specific type of secondary transport. The three transporters used in this work have different energetics; LacY and XylE are proton-coupled symporters but GlpT is an antiporter that uses P_i gradient to import G3P.

We agree with the reviewer that the role of a proton gradient is neglected. This was done on purpose to avoid diluting our main message: that lipid-protein interactions play an important role in regulating conformational transitions. As mentioned in our discussion, the motif that we have identified as the signature motif for these lipid-protein interactions is also present in uniporters, which suggest that this mechanism is independent from the presence of a gradient. Of course, this remains to be shown and we are currently developing new projects aiming at answering such questions.

Fig 1 caption should be improved:

- 1. “dHDX (red – more deuteration for G-W compared to WT) or negative dHDX (blue – less deuteration for G-W compared to WT)”***

This has now been modified.

- 2. Delete the fictitious D uptake curves (top right). Let the experimental data speak for themselves.***

We thank the reviewer for his/her comment. The addition of these curves was motivated by recurring questions that we had from the audience when presenting the work. HDX-MS is not (yet) a widely used technique in the community of membrane proteins structural biologists. This graph is aimed at familiarising the reader with the data presentation throughout the whole manuscript. Non-HDX experts who helped writing the paper found this helpful. At this stage, we would rather keep the figure as it is. However, we agree that this graph doesn't bring any new information and we don't mind removing these fictitious curves if the editor or the reviewers agree with this suggestion.

3. Figure and caption: simplify by not attempting to explain the experimental workflow.

This has now been modified.

4. Avoid using “accessibility” when discussing HDX data, instead the text should focus on H-bond stability as commonly done in the HDX literature.

We agree with the reviewer that the word accessibility is misleading and it has now been removed.

5. Top left: G-W and WT are both shown in the OF conformation; to casual readers it will not be clear how the protein conformation changes after mutation.

We have now changed the figure to show that the transporter is alternating between conformations in the absence of the mutation.

The authors are probably correct that the G-W mutation stabilizes the OF mutations. An alternative interpretation of Fig. 1b could be that the bulky Trp side chains causes disorder in its direct vicinity, thereby triggering the “red” regions in Fig. 1b. The authors may wish to address this possibility. Similarly, can the authors rule out that the mutations in the intracellular acidic sites (Section 2) cause local conformational destabilization (“red”), without shifts of the IF/OF equilibrium?

We thank the reviewer for this comment and want to clarify this aspect. For the three different transporters, this increase in uptake on one side is coupled to a decrease on the other side. The simplest explanation for this specific pattern is a shift in the IF/OF conformational equilibrium. It is unlikely that for three different transporters, the disorder caused by the mutation on one side would correlate with a structural stabilization on the other side. Furthermore, this increase in H/D exchange is observed at different locations on one side, some of them not in the vicinity of the mutation.

We had no *a priori* knowledge on the type of uptake patterns we would obtain by mutating the intracellular acidic sites residues and were very pleased to observe a pattern consistent with a change in the conformational equilibrium within the structural framework of the alternating access. It is worth noting that not all the mutations we carried out lead to this “binary” differential uptake pattern. In supplementary Fig.4, we can see that the mutations D27N on XyleE and E325Q on LacY caused a decrease in H/D exchange in their vicinity.

Another indirect evidence supporting our claim stems from additional molecular dynamics simulations. We carried out MD simulations of XyleE in the outward-facing conformation in PC bilayers for XyleE WT and the mutant E153Q. The water density plots presented here below show that the mutant starts to open on the intracellular

side and allows more water molecules to populate the cytosolic side (highlighted on the figure). No global conformational transition can be observed on such a short timescale but there is a definite trend towards a cytoplasmic opening, which comforts us that the mutation causes a conformational change towards an inward-facing conformation.

Figure 10. Water density plots of XylE WT (top panel) and XylE E153Q mutant (bottom panel) in DOPC bilayers.

Figure 7 and associated text: The transport cycle is unclear. (1) The proton is missing. (2) What is the evidence that PE binds *inside* the transport channel? (3) Clearly indicate intra- and extracellular side.

Both reviewers 2 and 3 pointed out that our model was confusing. We have now rewritten the paragraph (p.20 and 21) and remade figure 7 in the main text following their suggestions and comments.

Methods: “and D is the percentage of deuterium in the final labelling solution” Please be more specific. Is this the 30 min time point?

We believe that the word “final” was misleading. By “final” we mean the overall deuterium content present after mixing the protein sample with the labelling solution. This has now been clarified.

Also: The authors probably have a good reason why they did not use the standard approach of normalizing the HDX data to fully exchanged controls – this should be briefly explained.

Since our work only looks at relative uptake differences, there is no advantage of normalizing the data. We refer the reviewer to a review from John Engen which questions the use of this practice when no absolute deuterium uptake numbers are required (Wales, Eggertson et al. 2013). We have now added a sentence in the Methods section to explain our choice.

***** Additional Minor Comments:**

Methods: “TIP3P model was used for water.” This is stated twice

We thank the reviewer for spotting this mistake; this has now been corrected.

p. 13 typo: We performed two setS ...

This has now been changed. We thank the reviewer for reading the manuscript carefully.

Methods: “Simulations were performed at 310 K using Langevin dynamics with a damping constant of 0.5 ps⁻¹.” I don’t think this is a correct statement. Langevin dynamics usually refer to solvent-free simulations, whereas here the authors used explicit solvation by TIP3P water.

We respectfully disagree with the reviewer on this aspect. Langevin dynamics is a method in which random noise and friction terms are added to the forces calculated from the force field. While the original idea behind the method was indeed to represent the effect of the environment (e.g., solvent), the method can and is widely used in nearly all explicit-solvent simulations as well, to represent “additional” effects from the environment.

Introduction: “HDX reports the exchange of labile amide protons” change to “...HYDROGENS”

This has now been changed.

To streamline the text, consistently use “cytoplasmic side” or “intracellular side” but not a mix of both.

We have now replaced all the “cytoplasmic side” by “intracellular side”.

p. 20: “primarily found in the IF conformation, limiting substrate and/or proton leakage” This implies that OF would result in substrate and/or proton leakage ... but probably that’s not what the authors are trying to say here?

We refer the reviewer to our answer to reviewer 2, who pointed out the exact same sentence. This sentence was obviously misleading and the paragraph has now been rewritten (p.20 and 21).

Figure 5c: Why is the normalized count shown on a scale that goes up to 2.0? What do these units mean?

Count here is the count in a particular bin of the histogram i.e, bin height. The distribution data is normalized in such a way that the area under the curve is one, where area is the summation bin height*binwidth. These units are misleading and for this reason, we have now changed the scale legend to “arbitrary units”.

Introduction: Consider rewriting this sentence: “it enables conformational characterization of unlabelled, unmodified proteins” because the proteins detected by MS have been labeled (and hence modified). [it will be clear to many readers what the authors mean here, but still the wording is a bit misleading].

We thank the reviewer for his/her advice. We have now changed the sentence to “It enables conformational characterization of proteins in solution without requiring covalent modification of the target”.

Bibliographic references

- Bligh, E. G. and W. J. Dyer (1959). "A rapid method of total lipid extraction and purification." Can J Biochem Physiol **37**(8): 911-917.
- Chen, C. C. and T. H. Wilson (1984). "The phospholipid requirement for activity of the lactose carrier of Escherichia coli." J Biol Chem **259**(16): 10150-10158.
- Dijkman, P. M. and A. Watts (2015). "Lipid modulation of early G protein-coupled receptor signalling events." Biochim Biophys Acta **1848**(11 Pt A): 2889-2897.
- Dowhan, W. and M. Bogdanov (2011). "Lipid-protein interactions as determinants of membrane protein structure and function." Biochem Soc Trans **39**(3): 767-774.
- Entezami, A. A., B. J. Venables and K. E. Daugherty (1987). "Analysis of lipids by one-dimensional thin-layer chromatography." J Chromatogr **387**: 323-331.
- Findlay, H. E. and P. J. Booth (2017). "The folding, stability and function of lactose permease differ in their dependence on bilayer lipid composition." Sci Rep **7**(1): 13056.
- Hamai, C., T. Yang, S. Kataoka, P. S. Cremer and S. M. Musser (2006). "Effect of average phospholipid curvature on supported bilayer formation on glass by vesicle fusion." Biophys J **90**(4): 1241-1248.
- Hoi, K. K., C. V. Robinson and M. T. Marty (2016). "Unraveling the Composition and Behavior of Heterogeneous Lipid Nanodiscs by Mass Spectrometry." Anal Chem **88**(12): 6199-6204.
- Kaback, H. R. (2015). "A chemiosmotic mechanism of symport." Proc Natl Acad Sci U S A **112**(5): 1259-1264.
- Law, C. J., J. Almqvist, A. Bernstein, R. M. Goetz, Y. Huang, C. Soudant, A. Laaksonen, S. Hovmoller and D. N. Wang (2008). "Salt-bridge dynamics control substrate-induced conformational change in the membrane transporter GlpT." J Mol Biol **378**(4): 828-839.
- Nie, Y., I. Smirnova, V. Kasho and H. R. Kaback (2006). "Energetics of ligand-induced conformational flexibility in the lactose permease of Escherichia coli." J Biol Chem **281**(47): 35779-35784.
- Quistgaard, E. M., C. Low, F. Guettou and P. Nordlund (2016). "Understanding transport by the major facilitator superfamily (MFS): structures pave the way." Nat Rev Mol Cell Biol **17**(2): 123-132.
- Schneider, C. A., W. S. Rasband and K. W. Eliceiri (2012). "NIH Image to ImageJ: 25 years of image analysis." Nature Methods **9**(7): 671-675.
- Smirnova, I., V. Kasho, J. Y. Choe, C. Altenbach, W. L. Hubbell and H. R. Kaback (2007). "Sugar binding induces an outward facing conformation of LacY." Proc Natl Acad Sci U S A **104**(42): 16504-16509.
- Vitrac, H., M. Bogdanov and W. Dowhan (2013). "Proper fatty acid composition rather than an ionizable lipid amine is required for full transport function of lactose permease from Escherichia coli." J Biol Chem **288**(8): 5873-5885.
- Wadsater, M., S. Maric, J. B. Simonsen, K. Mortensen and M. Cardenas (2013). "The effect of using binary mixtures of zwitterionic and charged lipids on nanodisc formation and stability." Soft Matter **9**(7): 2329-2337.
- Wales, T. E., M. J. Eggertson and J. R. Engen (2013). "Considerations in the analysis of hydrogen exchange mass spectrometry data." Methods Mol Biol **1007**: 263-288.

REVIEWERS' COMMENTS:

Reviewer #1 (Remarks to the Author):

The response to my comments and changes make this worthy of publication.

Reviewer #2 (Remarks to the Author):

The authors have done a wonderful job in revising the manuscript; my compliments.

With respect to the "leak issues" raised by R2 and R3 the authors may want to read/cite the following paper, as it gives a framework on the different types of leak pathways in secondary active transporters.

Uncoupling in secondary transport proteins. A mechanistic explanation for mutants of lac permease with an uncoupled phenotype.

Lolkema JS, Poolman B.

J Biol Chem. 1995 May 26;270(21):12670-6.

Reviewer #3 (Remarks to the Author):

Accept.

Point-by-point answer to reviewers

Reviewer #1 (Remarks to the Author):

The response to my comments and changes make this worthy of publication.

Many thanks.

Reviewer #2 (Remarks to the Author):

The authors have done a wonderful job in revising the manuscript; my compliments.

Many thanks

With respect to the "leak issues" raised by R2 and R3 the authors may want to read/cite the following paper, as it gives a framework on the different types of leak pathways in secondary active transporters. "Uncoupling in secondary transport proteins. A mechanistic explanation for mutants of lac permease with an uncoupled phenotype. Lolkema JS, Poolman B. J Biol Chem. 1995 May 26;270(21):12670-6."

We have read the suggested article with interest. However, given the limitation on the number of references, we believe it should not be included in the final manuscript. Our work focuses on the role of lipids in the transport mechanism and, at this stage, we are not approaching the complex subject of proton leakage by secondary transporters.

Reviewer #3 (Remarks to the Author):

Accept.

We respectfully agree with the reviewer' suggestion.